# Effect of Heat Treatment on the Digestive Characteristics of Different Soybean Oil Body Emulsions

**DOI:** 10.3390/foods12152942

**Published:** 2023-08-03

**Authors:** Xufeng Yang, Luyao Zhou, Yingying Wu, Xiuzhen Ding, Wentao Wang, Dajian Zhang, Luping Zhao

**Affiliations:** 1College of Food Science and Engineering, Shandong Agricultural University, Tai’an 271018, China; 17863852660@163.com (X.Y.); 15194109917@163.com (L.Z.); yying1220@126.com (Y.W.); xzd@sdau.edu.cn (X.D.); wangwt@sdau.edu.cn (W.W.); 2Engineering and Technology Center for Grain Processing of Shandong Province, Tai’an 271018, China; 3State Key Laboratory of Crop Biology, Shandong Agricultural University, Tai’an 271018, China

**Keywords:** heat treatment, soybean oil body, protein hydrolysis, digestive characteristics, FFA release

## Abstract

Soybean oil body (SOB) emulsions were prepared using OBs extracted at pH 11.0 and pH 7.0. The pH 11.0-SOB comprised oleosins, whereas pH 7.0-SOB comprised extrinsic proteins and oleosins. All SOB emulsions were heated at 60–100 °C for 15 min. Heating may lead to the release of extrinsic proteins from the surface of pH 7.0-SOB due to heat-induced denaturation. The total proportion of α-helix and β-sheets gradually decreased from 77 (unheated) to 36.2% (100 °C). During stomach digestion, the extrinsic protein hydrolysis of heated pH 7.0-SOB emulsions was fast between 60 and 80 °C, and it then slowed between 90 and 100 °C; heating inhibited the oleosin hydrolysis of pH 7.0- and 11.0-SOBs. Heat treatment promoted aggregation and coalescence, and it resulted in increased particle sizes for all emulsions. Larger aggregates were found in heated pH 7.0-SOB emulsions, and larger oil droplets were found in heated pH 11.0-SOB emulsions. After intestinal digestion, the droplets of all SOB emulsions gradually dispersed, and particle sizes decreased. Different heating temperatures had lesser effects on particle sizes and microstructures. Lipolysis was affected by the extraction pH and heating. For pH 11.0-SOB emulsions, the FFA release tendency was greatly affected by the heating temperature, and heating to 80 °C resulted in the highest FFA release (74%). However, all pH 7.0-SOB emulsions had similar total FFA releases. In addition, the droplet charges of heated pH 7.0-SOB emulsions were lower than those of unheated pH 7.0-SOB emulsions in both the intestine and stomach phases; however, the charge changes in different pH 11.0-SOB emulsions showed the opposite tendency. This study will offer guidance regarding the application of SOB emulsions in food.

## 1. Introduction

Soybeans are widely cultivated as oil crops across the world. The protein and lipid contents of soybeans are 37.7–44.5% and 17.2–21.6%, respectively [1]. It is known that protein storage vacuoles act as storage organelles for soybean proteins, and soybean oil is stored in special organelles known as oil bodies (SOBs). A SOB is composed of a core of triglycerides (TAGs) coated with a monolayer of phospholipids that is embedded into the OB’s intrinsic proteins [2,3,4]. Intrinsic proteins include steroleosins, caleosins, and oleosins, and they play significant roles in resisting environmental pressure and maintaining the SOBs’ integrity [5]. Oleosins are small molecules with molecular weights of 15–26 kDa (mainly 24 kDa oleosins, 18 kDa oleosins, and 16 kDa oleosins). The extrinsic proteins include glycinin (11S: acid polypeptides, A; basic polypeptides, B), β-conglycinin (7S: α′, α, β), γ-conglycinin, Bd 30K, and P34 [6]. SOBs also contain phytosterol [7] and tocopherol [8], which are components required for healthy functioning. SOBs have great oxidation stability and dispersion stability due to their functional components and special structures. SOB emulsion preparation does not need the involvement of other emulsifiers or homogenization. It also conforms to food processing requirements related to safety and environmentally friendly practices.

Due to the need for food processing and storage, heat treatment is one of the most common techniques used in the field of food. Furthermore, heating is a common method used to increase the digestibility of proteins in various foods [9]. However, some studies found that heating could change the interfacial protein compositions and structures of OBs. Yan et al. (2016) found that extrinsic proteins/oleosins in soymilk were approximately 1.1 in unheated conditions [10]. After heating, the extrinsic proteins/oleosins were 0.72 (70 °C, 8 min), 0.43 (80 °C, 4 min), 0.34 (90 °C, 4 min), and 0.31 (100 °C, 4 min), respectively, indicating that the extrinsic proteins remained within the heated SOBs. As the SOBs were extracted at 60–100 °C for 30 min, the extrinsic proteins decreased from 65.29 to 4.74% as the heating temperatures increased [11]. In addition, the oleosin of the raw camellia OB was 13.3 kDa. The bands of 25–30 kDa and 55 kDa were determined based on the extended roasting time. The protein bands of the boiled OBs became shallow [12]. The study found that the secondary structure of the raw camellia OB interface proteins changed [12]. The random coil increased from 6% (raw camellia OB) to 13–18% (roasted seeds treated at 130 °C, for 5–10 min) or 13–23% (seeds boiled for 20–60 min). However, the β-sheet decreased from 25% (raw camellia OB) to 20% (roasting) or 5% (boiling). Fu et al. (2020) studied the effects of pre-heating treatments (including heating raw soymilk at 65 °C, 75 °C, and 85 °C for 30 min, as well as ultra-high temperature-treated dry soybeans at 110 °C, 120 °C, and 130 °C for 10 s) on SOBs [13]. They found that heating soymilk led to storage protein denaturing and partial unfolding, whereas ultra-high temperatures could lead to large amounts of soluble aggregates being present within oleosins and storage proteins emerging via disulfide bonds. Therefore, the interfacial protein structures and compositions of OBs were affected by heat treatment.

OBs could be considered as special protein-stabilized oil droplets because the entire surface was covered by proteins [14]. We considered the properties of the OBs’ surfaces to be similar to those of protein-stabilized emulsions. Some studies reported that heat treatment could affect the digestive characteristics of protein-stabilized emulsions and OB emulsions. Capuano et al. (2018) investigated the behavior of raw and roasted hazelnut OBs via in vitro lipid digestion [15]. They found that oleosins of raw hazelnut OBs were hydrolyzed to produce an 8.0-dalton pepsin-resistant peptide after gastric digestion, and the free fatty acid (FFA) releases of raw and roasted hazelnut OBs were 31% and 45%, respectively. Ding et al. (2020) studied the digestive characteristics of SOBs that were extracted and heated (95 °C, 15 min), along with non-heated raw soybeans, via in vitro gastrointestinal digestion [16]. It is worth noting that the aggregation statuses of droplets for both heated and non-heated SOBs are similar in the stomach phase. Conversely, the FFA release rate of heated SOBs was higher (23.01%) than that of non-heated SOBs (21.05%). Ye et al. (2020) studied the digestive behaviors of whey protein-stabilized emulsions treated using heat treatment, and they found that the heated whey protein-stabilized emulsion had a higher degree of protein hydrolysis [17]. During in vitro intestinal digestion, the FFA releases of the heated emulsion and unheated emulsion were 65% and 52%, respectively. Therefore, we postulated that heat treatment changed the structures and compositions of OB interfacial proteins, which could affect the digestive regularity of OB emulsions. However, the digestive characteristics of SOB emulsions treated at different heating temperatures were not systematically investigated. In addition, we previously reported that the digestive characteristics of SOBs were different [18]. Therefore, the digestive characteristics of pH 7.0- and 11.0-SOBs that were heat treated needed to be studied.

In this study, pH 7.0- and pH 11.0-SOB emulsions were each treated at 60 °C, 70 °C, 80 °C, 90 °C, and 100 °C. The digestive regularities of pH 7.0- and pH 11.0-SOB emulsions were researched in the gastrointestinal stage, including protein hydrolysis, zeta potentials, particle sizes, microstructures, and FFA releases. This study will offer theoretical guidance for the application of SOB emulsions in food, such as the delivery of functional nutrients and dietary lipid.

## 2. Materials and methods

### 2.1. Materials

We bought soybean seeds (Zhonghuang 37) from the local market (Tai’an, China). Protein markers (14.4–97.4 kDa) were bought from Solarbio (Beijing Solarbio Science and Technology Co., Ltd., Beijing, China). Pancreatin, pepsin, Fast Green and Nile Red, and bile salts were bought from Sigma-Aldrich (St. Louis, MO, USA). All chemical reagents were analytical grade.

### 2.2. Extractions of pH 11.0-SOB and pH 7.0-SOB

The SOBs were extracted using the modified methods described by Han et al. (2020) [19]. Soybean seeds (50 g) were washed with deionized (ID) water to remove dust and soaked in fresh ID water (300 g) for 16 h at 4 °C in a refrigerator. The swollen seeds were collected and washed three times with fresh ID water. The pre-cooled (4 °C) ID water was mixed with the swollen seeds (ID water/seeds = 9/1, *w/w*), and the mixture was ground via a crushing machine (BLST4090-073, Qster) at 22,000 r/min for 2 min. Then, four layers of gauze were used to filter the resulting slurry. Sucrose (62.5 g) was added to the filtrate (312.5 g) and mixed well. The mixture was divided into two parts, and their pHs were adjusted to 11.0 and 7.0, respectively, using 0.2–2 mol/L KOH and HCl solutions. The mixtures were centrifuged at 43,667× *g* and 4 °C for 30 min. The floats were collected from the upper layer and mixed again with fresh ID water. The pHs of the mixtures were adjusted to 11.0 and 7.0, respectively, and they were centrifuged under the same conditions. The above steps were repeated twice. The obtained SOBs were called pH 11.0-SOB and pH 7.0-SOB, respectively.

### 2.3. Emulsion Preparation and Heat Treatment

SOBs (8.0 g) were dispersed into the ID water to prepare the SOB emulsions (5%, dry basis). All emulsions were sheared (3000 r/min, 2 min) using a dispersion machine (T25 Digital Package 2, German IKA Company, Königswinter, Germany). The pH 7.0- and pH 11.0-SOB emulsions were heated at 60 °C, 70 °C, 80 °C, 90 °C, and 100 °C for 15 min and immediately cooled. An unheated SOB (4 °C) was used as the control. Some samples were centrifuged (11,102× *g*) again, and the floats were analyzed using Tricine-SDS-PAGE. NaN_3_ (0.2%, *w/w*) was added into the other SOB emulsions, which were immediately used for further analysis.

### 2.4. In Vitro Gastrointestinal Digestion of SOBs

The digestive model was designed according to the method previously mentioned by Yang et al. (2022) [18]. Each SOB emulsion (80.0 g) was mixed with 20 mL of simulated gastric fluid SGF (the SGF contained 0.43 mol/L of NaCl). The pH of each mixture was controlled to pH 4.0 using HCl (0.2–2 mol/L), and pepsin (0.68 mg/mL) was added to each mixture. The simulated stomach digestion occurred under the conditions of pH 4.0 and 37 °C for 2 h.

When simulated stomach digestion was completed, the pH of each mixture was adjusted to 7.0 with KOH (0.2–2 mol/L). The simulated intestinal fluid (SIF) was composed of CaCl2 (0.3 mol/L), KCl (25 mmol/L), and NaCl (0.75 mol/L). Bile salts (0.62 g) were added into the prepared SIF (57.6 mL). Then, pancreatin was added into the solution, and the final concentration was 6.35 mg/mL. This solution simulated intestinal digestion at pH 7.0 and 37 °C, which lasted for 3 h.

### 2.5. Zeta Potential

Samples were collected at 120 min (termed G120) in the stomach phase, cooled, and diluted 400 times using citric acid solution (20 mmol/L, pH 4.0). In addition, samples were collected at 180 min (termed I180) in the intestine phase and cooled. The initial emulsions and all samples were also diluted for the number of times described above using phosphate solution (20 mmol/L, pH 7.0). Zeta potentials were carried out using a Laser Nano-ZS Particle Analyzer (Malvern Instruments Ltd., London, UK) at 25 °C [19,20].

### 2.6. Droplet Size Analysis

Samples from stomach and intestinal digestion were diluted 1000 times using a citric acid solution (20 mmol/L, pH 4.0) and a phosphate solution (20 mmol/L, pH 7.0), respectively. The particle sizes were determined using a Laser Nano-ZS Particle Analyzer (Malvern Instruments Ltd., London, UK) at 25 °C.

### 2.7. Tricine-SDS-PAGE

Tricine-SDS-PAGE was conducted according to the method of Zhao et al. (2016) [21], albeit with modifications. The concentrations of stacking and separating gels were 4% and 16%, respectively. The protein concentrations of all samples were adjusted to 1.5 mg/mL via mixing with Tricine-SDS-PAGE sample buffer. β-mercaptoethanol (2%, *v/v*) was added to all samples, and the mixed solutions were heated at 100 °C for 5 min. Then, 8 μL of each sample was loaded into the sample well, and samples were electrophoresed at a constant voltage of 30–35 mV until all samples entered into the stacking gel, and the constant voltage was then changed to 100 mV until complete. The gel was stained using Coomassie Brilliant Blue G-250. The intensities of different protein bands were analyzed via Image Lab 3.0 Software (Bio-Rad, Hercules, CA, USA).

### 2.8. Microscope Observation

Microstructures were observed using a Confocal Laser Scanning Microscope (CLSM) 880 (Carl Zeiss, Jena, Germany). The oils and proteins were stained using 0.02% (*w/v*) Nile Red dissolved in 10% (*v/v*) acetone and 0.1% (*w/v*) Fast Green aqueous solution, respectively. Samples (200 μL) were stained with Nile Red (20 μL) and Fast Green (25 μL) and stored with restricted lighting for 10 min [22,23]. The excitation wavelengths of Nile Red and Fast Green were 488 nm and 633 nm, respectively, which were detected at 580–620 nm and 651–710 nm. Samples were observed with 43× oil lenses. Zeiss Zen 2012 Software (Jena, Germany) was used to analyze all images.

### 2.9. Free Fatty Acid (FFA) Release

During intestinal digestion, pH-STAT automatic potentiometric titration apparatus was used to monitor the consumed amount of KOH (0.1 mol/L) [24,25]. FFA release was calculated on the basis of KOH consumption, as per the below equation:(1)$$FFA\;(\%)=\frac{V_{KOH\;}{\times \;m\;}_{KOH}\;\times \;M}{M_{lipid\;}\times \;2}\;\times \;100$$

*V_KOH_* and *m_KOH_* represent the consumed volume and content of KOH, respectively; *M* represents the molecular weight of SOBs, i.e., 876.56 g/moL; and *M_lipid_* represents the weight of TAGs.

### 2.10. Statistical Analysis

Experiments were performed on 3 separate occasions in triplicate using freshly prepared samples. The data were presented as the means with standard deviations. Statistical analysis was performed using SPSS statistics version 25 following an ANOVA model (IBM Inc., Armonk, NY, USA). A confidence level of *p* < 0.05 was considered to be statistically significant.

## 3. Results and Discussion

### 3.1. Protein Compositions and Structures of pH 7.0- and pH 11.0-SOB Emulsions

The protein compositions of unheated and heated pH 7.0- and pH 11.0-SOB emulsions are shown in Figure 1a,c. The unheated pH 7.0-SOB had oleosins (24 kDa, 18 kDa, and 16 kDa) and extrinsic proteins (main 7S: α, α′, and β; 11S: A, A3 and B). However, the unheated pH 11.0-SOB only had oleosins. Compared to unheated SOBs, the protein compositions of the pH 7.0- and pH 11.0-SOB emulsions heated at 60 °C–100 °C showed no significant changes. In order to determine the proteins bound to SOBs, the heated pH 7.0- and pH 11.0-SOB emulsions were centrifuged, and their protein compositions are shown in Figure 1b,d. It was found that all 7S and 11S were released from the pH 7.0-SOB heated to more than 60 °C; however, oleosins were still bound to the heated pH 7.0- and 11.0-SOB. Oleosins formed “T” shapes on the SOB surfaces. The N-terminal and C-terminal regions of the peptide chain were exposed to the outside, and the hydrophobic region in the middle of the peptide chain was embedded firmly inside of the SOB [5]. Therefore, extrinsic proteins were released at ≥60 °C.

The Raman spectra of the pH 7.0-SOB and the secondary structures of the interfacial proteins are shown in Figure 2. The peaks featured in the analysis of the protein structure were referred to as amide I (1600–1700 cm^−1^) and amide III (1230–1300 cm^−1^). In order to reveal the protein structure of the pH 7.0-SOB, we quantitatively fitted amides I and III via the Raman spectrum (Figure 2b). The secondary protein structure of the unheated pH 7.0-SOB was mainly made up of 45.6% α-helix and 31.4% β-sheet. However, the α-helix gradually decreased from 45.6 to 19.5% with increases in the heating temperatures, and the random coil increased from 18 (60 °C) to 37.5% (100 °C). In addition, the total proportion of α-helix and β-sheet decreased from 72 (unheated) to 36.2% (100 °C). This result indicated that heating could gradually change the protein’s structure from an ordered structure to a disordered structure, which explained the release of 7S and 11S from the pH 7.0-SOB surface (Figure 1b). Some studies reported that α′/α and β were separated from 7S, and A and B were separated from 11S by heating raw soybean milk, resulting in 7S and 11S being released from the SOBs [26,27].

### 3.2. Protein Hydrolysis in Stomach Digestion

Protein hydrolysis was investigated in the stomach digestion phase, and the results are shown in Figure 2 and Figure 3. All protein bands of each SOB emulsion became gradually shallower and produced low molecular bands with the extension of the digestive time, which indicated that proteins were gradually hydrolyzed and produced small molecular peptides.

The effects of heat treatment on extrinsic protein and oleosin hydrolysis were greatly different for pH 7.0-SOB emulsions. The extrinsic proteins of the heated pH 7.0-SOB emulsions were more rapidly hydrolyzed than those of the unheated pH 7.0-SOB after 120 min (Figure 3), indicating that the released extrinsic proteins were more easily hydrolyzed than the extrinsic proteins bound to the unheated pH 7.0-SOB. The hydrolysis of extrinsic proteins became faster from 60 to 80 °C, before slowing from 90 to 100 °C. Fu et al. (2020) reported that the 7S and 11S denaturation temperatures were about 70 °C and 90 °C, respectively [13]. Under their denaturation temperatures, 7S and 11S were partially denatured, which probably exposed the active groups and enzyme binding sites inside the molecule. However, 7S and 11S were completely denatured at 90 °C–100 °C, resulting in the thermal aggregation of extrinsic proteins, and the exposed enzyme binding sites were covered again [17]. Compared to the unheated pH 7.0-SOB emulsion, heat treatment inhibited oleosin hydrolysis. For example, 24 kDa of oleosin in the unheated pH 7.0-SOB was completely hydrolyzed at 5 min, and hydrolysis of 18 kDa of oleosin required 30 min; for 100 °C, 24 kDa of oleosin was completely hydrolyzed at 30 min, while 18 kDa of oleosin was still not completely hydrolyzed at 120 min. In addition, oleosin hydrolysis rates showed a tendency to decrease with the increases in the heating temperatures for pH 7.0-SOB emulsions. Interestingly, the amount of 16 kDa first showed an increasing tendency and then decreased, as 24 kDa of oleosin was hydrolyzed and produced 16 kDa of peptide [4,28]. Therefore, heating greatly affected the protein hydrolysis of pH 7.0-SOB emulsions.

Heat treatment also affected oleosin hydrolysis for pH 11.0-SOB emulsions. For the unheated pH 11.0-SOB emulsion, 24 kDa of oleosin was completely hydrolyzed at 5 min, and 18 kDa of oleosin was hydrolyzed at 20 min. At 100 °C, most of the 24 kDa of oleosin was hydrolyzed at 60 min, and most of the 18 kDa of oleosin was hydrolyzed at 120 min. It was suggested that heating inhibited oleosin hydrolysis. In addition, the oleosin hydrolysis rate decreased with the increases in the heating temperatures, which was similar to the result for the pH 7.0-SOB (Figure 4). Small pepsin-resistant peptides (marked in black) were produced from oleosins in the stomach phase in all pH 11.0-SOB emulsions.

### 3.3. Zeta Potential

It was shown that the initial pH 7.0- and pH 11.0-SOB emulsions had negative charges, and their zeta potentials were positive in the stomach phase and changed to negative during intestinal digestion (Figure 5), which occurred due to pH values of 4.0 and 7.0 in the gastric and intestinal environments, respectively. Before digestion, the zeta potentials of the unheated pH 7.0- and 11.0-SOB were −21.7 mV and −15.67 mV, respectively, which was due to the extrinsic proteins bound to the pH 7.0-SOB. This result was consistent with Yang et al. (2022), and they reported that the zeta potential of SOB emulsions increased with the increases in the extraction pHs from 5.0 to 11.0 [18]. In addition, the zeta potentials of pH 7.0-SOB emulsions decreased from −18.9 mV at 60 °C to −22.83 mV at 70 °C, before increasing to −11.15 mV at 100 °C (Figure 5a), which might be relative to structures of the interfacial proteins. Heating temperatures (60 °C–100 °C) had no effects on the zeta potentials of pH 11.0-SOB emulsions because heated pH 11.0-SOB emulsions had the same protein compositions (Figure 1c,d).

After stomach digestion for 120 min, the zeta potentials of heated pH 7.0-SOB emulsions decreased significantly compared to the unheated pH 7.0-SOB (Figure 5a). This discrepancy was attributed to the faster hydrolysis of extrinsic proteins in heated pH 7.0-SOB emulsions (Figure 3). Zeta potentials decreased with the increase in the heating temperature from 60 to 80 °C, before increasing with the increase from 80 to 100 °C. These changes were consistent with the regularity of extrinsic protein hydrolysis in different heated pH 7.0-SOB emulsions (Figure 3). It was revealed that extrinsic protein hydrolysis had important effects on the zeta potentials of pH 7.0-SOB emulsions. In addition, the zeta potentials of unheated pH 11.0-SOB emulsions were lower than those of heated pH 11.0-SOB emulsions after stomach digestion. This result was attributed to the fact that more oleosins were hydrolyzed for unheated pH 11.0-SOB emulsions. The zeta potentials of pH 11.0-SOB emulsions experienced no significant changes with the increase in temperature from 60 to 100 °C. After intestinal digestion for 180 min, the zeta potentials of all heated pH 7.0-SOB emulsions were slightly higher than those of unheated pH 7.0-SOB emulsions, and the zeta potentials were highest when heated at 80 °C. However, different heated pH 11.0-SOB emulsions had similar zeta potentials, which were significantly lower than those of unheated pH 11.0-SOB emulsions. McClements and Li. (2010) reported that small molecular substances (such as FFAs, bile salts, and peptides) were absorbed via the droplet interface [29], which increased droplet charges. It is possible that different amounts of small molecular substances were absorbed via the droplet interface.

As a result, the zeta potential of the unheated pH 11.0-SOB emulsion was higher than that of the pH 7.0-SOB with extrinsic proteins before digestion, and the heating temperature had a greater effect on the zeta potential of the pH 7.0-SOB. Moreover, the heated pH 7.0-SOB emulsions had fewer droplet charges than those of the unheated pH 7.0-SOB in both the intestine and stomach phases; however, the charge changes in pH 11.0-SOB emulsions showed the opposite tendency. The heating temperatures (60–100 °C) had no obvious influence on the zeta potentials of pH 11.0-SOB emulsions.

### 3.4. Particle Size

Before digestion, the average particle size of the unheated pH 7.0-SOB emulsion was 658 nm, and the particle sizes of heated pH 7.0-SOB emulsions first decreased to 459 nm (60 °C), before remaining constant at about 327 nm (70–100 °C). This result occurred due to extrinsic proteins released from the pH 7.0-SOB surface via heating (Figure 1a,b). Conversely, there were no significant changes in the particle sizes of unheated and heated pH 11.0-SOB emulsions because the unheated and heated pH 11.0-SOB emulsions had the same protein composition (Figure 1c,d).

It was shown that the particle sizes of pH 7.0- and pH 11.0-SOB emulsions increased significantly in the stomach phase (Figure 6). The reason for this increase was that the pH of gastric fluid was close to the isoelectric points of SOB interfacial proteins, which further reduced electrostatic repulsion and droplet aggregation [30]. Moreover, a coalescence of droplets occurred due to the hydrolysis of extrinsic proteins and oleosins [31]. Compared to the unheated pH 7.0-SOB, the particle sizes of all heated pH 7.0-SOB emulsions were larger, and this result might be attributed to the fact that more aggregates formed due to weak electrostatic repulsion (Figure 5a). However, the particle sizes of heated pH 11.0-SOB emulsions with more positive charges were higher than those of unheated pH 11.0-SOB emulsions with fewer positive charges. It was possible that the droplet coalescence had greater effects on particle size than on aggregation. When heating temperature increased from 60 to 100 °C, the droplet sizes of pH 11.0-SOB emulsions showed a tendency to decrease. This result might be attributed to the fact that oleosin hydrolysis became slower, and the degree of droplet coalescence decreased with the increase in the heating temperature. Compared to stomach digestion, the particle sizes of all pH 7.0-and pH 11.0-SOB emulsions significantly decreased due to droplets having more negative charges after intestinal digestion for 180 min. There were no regular changes in particle sizes for pH 7.0-SOB emulsions in the intestinal phase. Compared to the particle sizes in unheated emulsions, heating at 60 °C and 70 °C, and 80–100 °C decreased the particle sizes of pH 11.0-SOB emulsions. Therefore, the heating temperatures had more effect on particle sizes in the stomach phase than in the intestinal phase.

### 3.5. Microstructures

The droplets were observed using CLSM, and oil phases and proteins were marked with red and green fluorescent dyes (Figure 7). Before digestion, the droplets of all initial pH 11.0- and 7.0-SOB emulsions dispersed well. During the stomach phase, the aggregation degree of droplets in pH 7.0-SOB emulsions first increased, before decreasing with the increases in the heating temperature, and more and larger aggregates could be observed in pH 7.0-SOB emulsions heated at 80 °C. This result was consistent with the particle size analysis (Figure 6a). In addition, small oil droplets were present in pH 7.0-SOB emulsions, suggesting that slight droplet coalescence occurred due to oleosin and extrinsic protein hydrolysis. Aggregates and large oil droplets occurred in heated pH 11.0-SOB emulsions during stomach digestion. These results revealed that aggregation and coalescence were the reasons for the increase in particle sizes of all emulsions in the stomach phase (Figure 6), and heat treatment resulted in larger oil droplets in pH 11.0-SOB emulsions and larger aggregates in pH 7.0-SOB emulsions. These aggregations and the coalescence induced by heating might increase steric hindrance, and oleosin hydrolysis was inhibited (Figure 3 and Figure 4). After intestinal digestion, the droplets gradually dispersed, the particle sizes of all SOB emulsions showed decreasing tendencies, and microstructures showed no obvious changes in the different SOB emulsions. Thus, the heating temperatures and extraction pHs of SOB emulsions had more influence on microstructures in the stomach phase than those in intestinal digestion.

### 3.6. The Release of FFAs

During intestinal digestion, TAGs were hydrolyzed using pancreatin, and the total FFA release increased gradually with increasing digestive time (Figure 8). The FFA release rate of unheated and heated pH 7.0-SOB emulsions first experienced a rapid increase, followed by a flat curve. For the pH 7.0-SOB emulsion, the FFA release rate increased with increases in the heating temperatures for 0–10 min, and the total FFA release was similar (about 70%) at 180 min. In addition, FFA release was greatly affected by the heating temperatures in the pH 11.0-SOB emulsion. For unheated and heated (60 °C and 70 °C) pH 11.0-SOB emulsions, the FFA release curve showed a transient lag phase during initial digestion, and the FFA content increased with prolonged digestive time at similar rates. The FFA release of the pH 11.0-SOB emulsion was heated at 80 °C, 90 °C, and 100 °C; it first rapidly increased, before continuing to increase more slowly. Finally, the total FFA releases were 60%, 65.25%, 64.5%, 74%, 62.1%, and 70.3% for pH 11.0-SOB emulsions, which were unheated and heated at temperatures in the range 60–100 °C, respectively. Moreover, the FFA release rate of pH 11.0-SOB emulsions increased with increases in the heating temperatures in a period of 0 to 40 min. These results were consistent with those of Chen et al. (2022), who found that the FFA release of unheated camellia OBs was only 41.7%, while boiling treatment for 60 min increased the FFA release to 57.4%, as well as the rate of lipolysis, during the initial digestion stage [12]. Therefore, the extraction pH affected lipolysis, and the heating temperatures promoted FFA release during the initial digestion stage and had more effects on the lipolysis of pH 11.0-SOB emulsions.

## 4. Conclusions

In this study, we prepared emulsions using SOBs extracted at pHs of 7.0 and 11.0 and discussed the effects of heat treatment (60–100 °C) on their digestive characteristics. It was found that the pH 11.0-SOB had only oleosins (16 kDa, 18 kDa, and 24 kDa), while the pH 7.0-SOB had extrinsic proteins (7S and 11S) and oleosins. Obviously, extrinsic proteins were released from pH 7.0-SOBs heated at 60–100 °C due to heat-induced denaturation. The total proportion of α-helix and β-sheet in the unheated pH 7.0-SOB was 77%, while the total proportion of α-helix and β-sheet decreased from 61.1 (60 °C) to 36.2% (100 °C). The heating temperatures had no significant effects on particle sizes and microstructures, and they had a greater effect on the zeta potentials of pH 7.0-SOBs than those of pH 11.0-SOBs before digestion. In stomach digestion, (1) the extrinsic protein hydrolysis of pH 7.0-SOB emulsions gradually accelerated from 60 to 80 °C, before slowing from 90 to 100 °C, and heat treatment inhibited oleosin hydrolysis for both the pH 7.0- and 11.0-SOB. Next, (2) the particle sizes increased for all emulsions, and heat treatment resulted in larger oil droplets in pH 11.0-SOB emulsions and larger aggregates in pH 7.0-SOB emulsions. Finally, (3) heat treatment decreased and increased the positive charges of SOB emulsions extracted at pH 7.0 and 11.0, respectively. After intestinal digestion, (1) the droplets gradually dispersed, and the particle sizes of all SOB emulsions decreased. Next, (2) heated pH 7.0–SOB emulsions had more negative charges than unheated emulsions, while the charge changes in pH 11.0-SOB emulsions showed the opposite trend. Finally, (3) for all pH 7.0-SOB emulsions, the total FFA releases were similar (about 70%) at 180 min, but different temperatures greatly affected the FFA releases of pH 11.0-SOB emulsions, and heating at 80 °C resulted in the highest FFA releases (74%). FFA release was improved in tandem with the increases in the heating temperatures during the initial digestion stage. Therefore, extraction pH and heat treatment had important effects on the digestive characteristics of SOB emulsions, which will provide guidance for selecting proper extraction pHs and heating temperatures for the application of SOB emulsions in the field of food.

## Figures and Tables

**Figure 1 foods-12-02942-f001:**
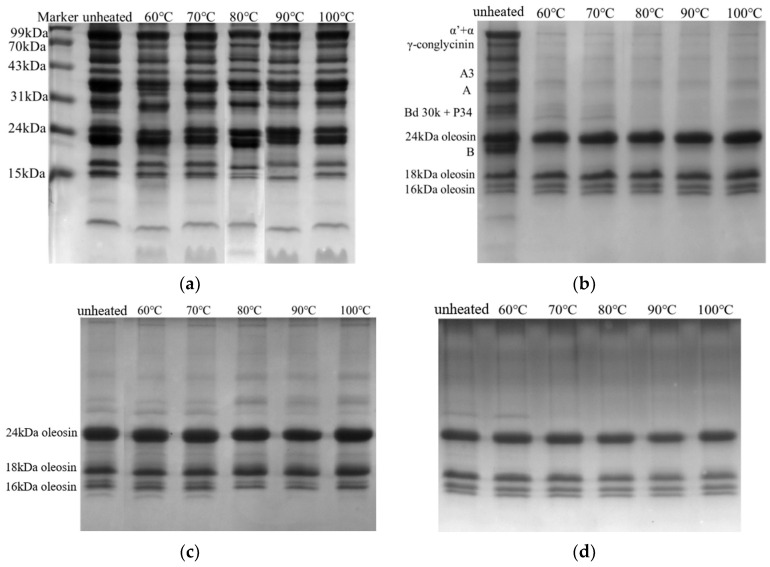
Tricine–SDS–PAGE protein profiles of pH 7.0-SOB (**a**,**b**) and 11.0-SOB (**c**,**d**) treated using different heat treatments. (**a**,**c**) represents heated SOBs; (**b**,**d**) represents the floats obtained by centrifuging heated SOB emulsions.

**Figure 2 foods-12-02942-f002:**
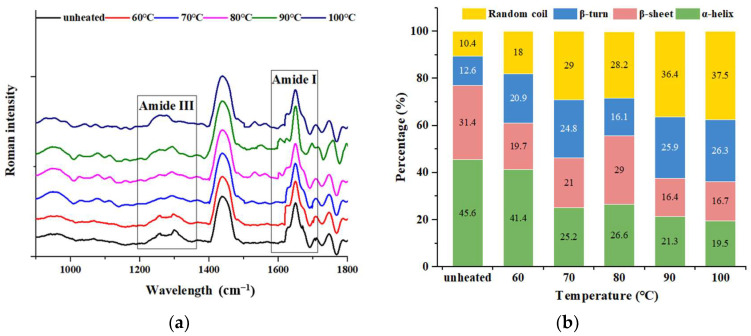
Confocal Raman spectra of SOB interface proteins (**a**), and the secondary structure determined via curve-fitting of Raman spectra (**b**).

**Figure 3 foods-12-02942-f003:**
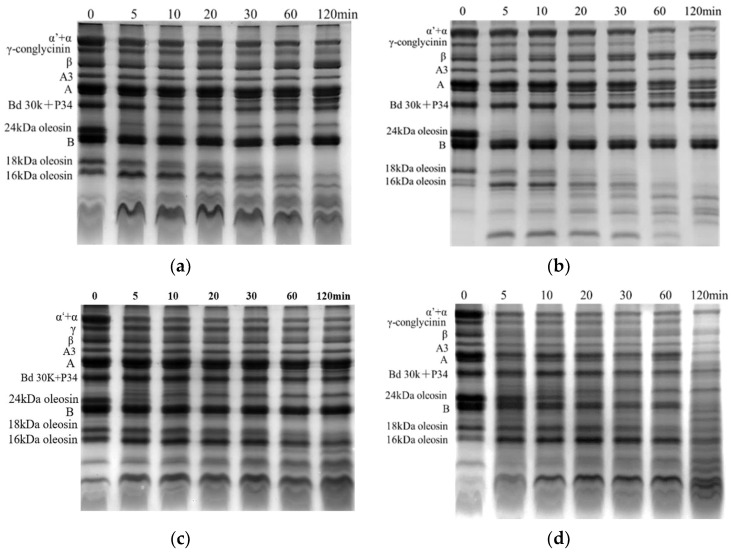
Tricine-SDS-PAGE of pH 7.0-SOB during the stomach phase. (**a**–**f**), unheated, 60 °C, 70 °C, 80 °C, 90 °C, 100 °C.

**Figure 4 foods-12-02942-f004:**
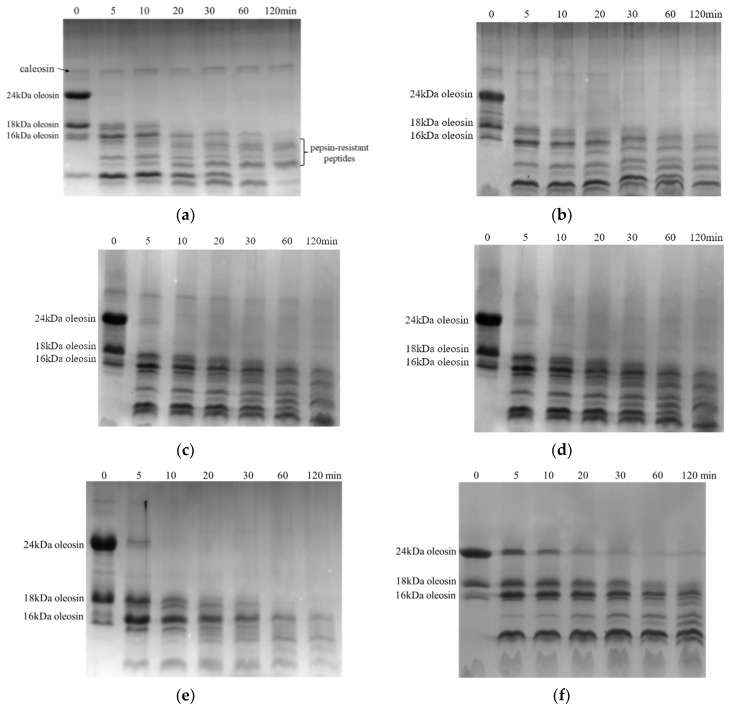
Tricine-SDS-PAGE of pH 11.0-SOB during the stomach phase. (**a**–**f**), unheated, 60 °C, 70 °C, 80 °C, 90 °C, and 100 °C.

**Figure 5 foods-12-02942-f005:**
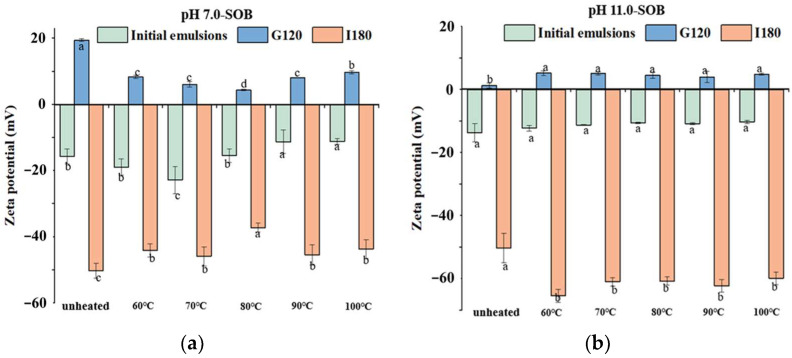
Zeta potential of pH 7.0-SOB (**a**) and 11.0-SOB (**b**) emulsions treated using different heat treatments during gastrointestinal digestion. Different small letters represent significant differences (*p* < 0.05) in the same digestion stage. G120 represents SOB emulsions incubated for 120 min during the stomach phase. I180 represents SOB emulsions incubated for 180 min during the intestinal phase. The following figure is similar.

**Figure 6 foods-12-02942-f006:**
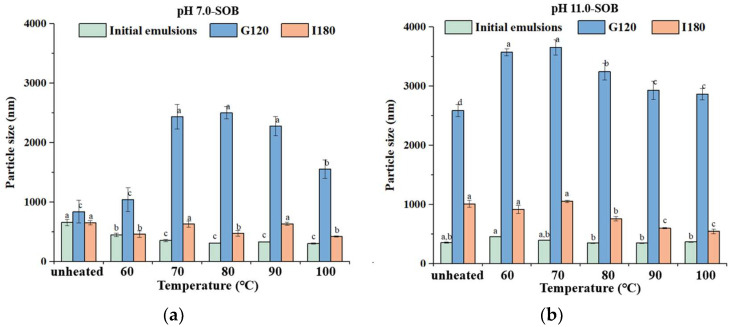
Particle size of pH 7.0-SOB (**a**) and 11.0-SOB (**b**) emulsions treated at different treatments during gastrointestinal digestion. Different small letters represent significant differences (*p* < 0.05) in the same digestion stage.

**Figure 7 foods-12-02942-f007:**
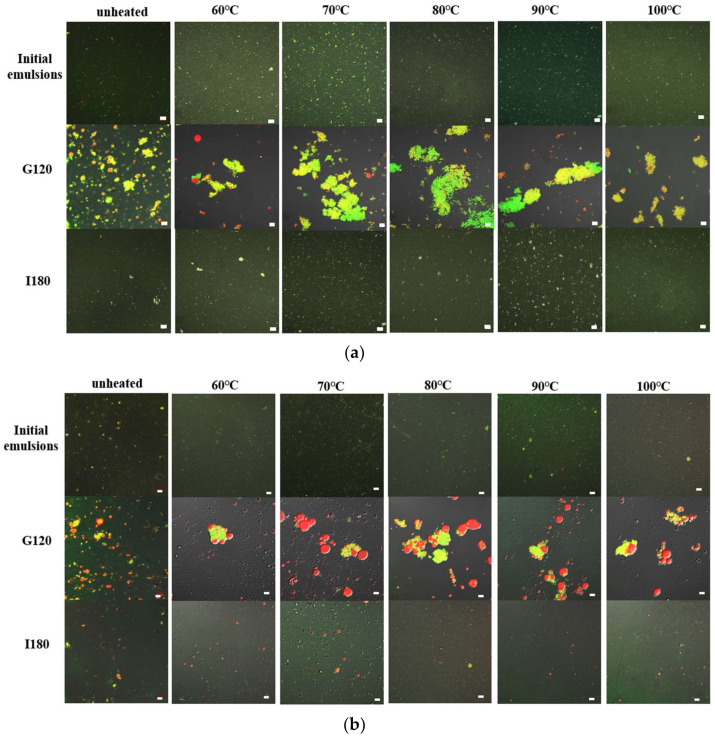
Confocal scanning microscopic images of pH 7.0-SOB (**a**) and 11.0-SOB (**b**) emulsions treated using different treatments during gastrointestinal digestion. Oil was stained using Nile Red (red), and the proteins were stained using Nile Blue (green).

**Figure 8 foods-12-02942-f008:**
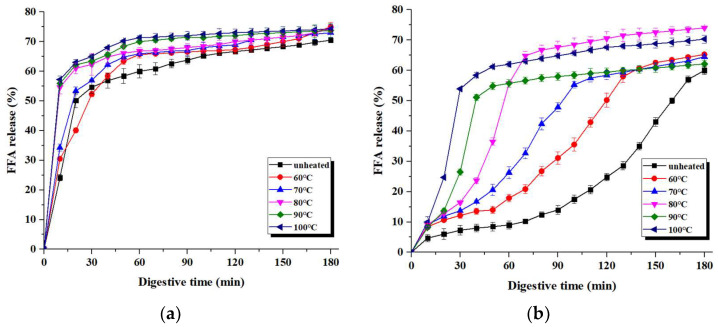
The FFA release of pH 7.0-SOB (**a**) and 11.0-SOB (**b**) emulsions treated using different treatments during intestinal digestion.

## Data Availability

The datasets generated for this study are available on request to the corresponding author.

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
