# Peer review of "Effect of Heat Treatment on the Digestive Characteristics of Different Soybean Oil Body Emulsions"

_foods, 2023, doi:10.3390/foods12152942_

Round 1
Reviewer 1 Report
Dear Authors,
The subject of the manuscript is very interesting. Many systems in food are emulsions. Soy, on the other hand, has a very well-balanced protein with high nutritional potential.
The research presented here makes it possible to obtain a clean-label system, without unnecessary additives, in terms of ecology and food safety. These are current trends in food production and research.
The research is very well planned and conducted in accordance with the rules, and the results are promising in terms of applicability.
There are a few editorial errors in the paper that should be corrected, such as the literature list.
Author Response
Point 1:
The subject of the manuscript is very interesting. Many systems in food are emulsions. Soy, on the other hand, has a very well-balanced protein with high nutritional potential.
The research presented here makes it possible to obtain a clean-label system, without unnecessary additives, in terms of ecology and food safety. These are current trends in food production and research.
The research is very well planned and conducted in accordance with the rules, and the results are promising in terms of applicability.
There are a few editorial errors in the paper that should be corrected, such as the literature list.
Response 1: Thank you for your comments, and we have been checked and some changes within the text are highlighted with a red font.
Reviewer 2 Report
This is interesting research on the effect of heat treatment on digestive characteristics of different
soybean oil body emulsions. However, the manuscript would surely benefit from revision by the authors as some grammatical errors are present in the text. I have indicated some of them by yellow highlighting them in the original manuscript.
-Revise the format style of all references within the text based on guidelines
-Line 43-46: Change the tense of the verbs to the present
-Line 48: Do not need
-Line 87: This abbreviation is written out in full the first time used. Therefore, the full name should be deleted here.
-Why was the dilution factor for zeta potential measurement differ from particle size measurement?
-Please clearly explain the reasons for positive and negative charges in the stomach and intestinal conditions.
-As you know, the high negative electrical charges are an indication of the high stability of emulsions. Please first explain the reason for this high zeta potential in intestinal conditions and then add more explanation for the advantages /disadvantages of this super stability in the intestinal.
Line 331: mentions that “After intestinal digestion, particle sizes of all SOB emulsions decreased”. However, the micrographs represent the aggregate particles.
-Update the old references

The English wiring should be double-checked to revise some grammatical errors (some of them are highlighted in the text)
Reviewer 3 Report
Effect of Heat Treatment on Digestive Characteristics of Different Soybean Oil Body Emulsions
Overall, the study presents valuable insights into the effect of heat treatment on the digestive characteristics of different SOB emulsions. The manuscript needs minor modifications.
Comments:
Ø At present, the Similarity Index of the manuscript is 31%. It should be less the 15% or as per Journal guidelines
Ø Why the pH 7.0 and 11.0 selected for the study?
Ø Line 100
This research will provide theoretical guidance for the application of SOB emulsions in the food field.
Needs more elaboration on what type of application
Ø It would be beneficial if the authors could address any limitations or potential sources of error in the study. Discussing the limitations would help readers understand the scope
Ø There are some English grammar problems in the manuscript which needs to be corrected
There are some English grammar problems in the manuscript which needs to be corrected
